

# The adder (*Vipera berus*) in Southern Altay Mountains: population characteristics, distribution, morphology and phylogenetic position

Shaopeng Cui[1,2], Xiao Luo[1,2], Daiqiang Chen[1,2], Jizhou Sun[3], Hongjun Chu[4,5], Chunwang Li[1,2] and Zhigang Jiang[1,2]

[1] Key Laboratory of Animal Ecology and Conservation Biology, Institute of Zoology, Chinese Academy of Sciences, Beijing, China
[2] University of Chinese Academy of Sciences, Beijing, China
[3] Kanas National Nature Reserve, Buerjin, Urumqi, China
[4] College of Resources and Environment Sciences, Xinjiang University, Urumqi, China
[5] Altay Management Station, Mt. Kalamaili Ungulate Nature Reserve, Altay, China

Corresponding authors
Chunwang Li, licw@ioz.ac.cn
Zhigang Jiang, jiangzg@ioz.ac.cn

## ABSTRACT

As the most widely distributed snake in Eurasia, the adder (*Vipera berus*) has been extensively investigated in Europe but poorly understood in Asia. The Southern Altay Mountains represent the adder's southern distribution limit in Central Asia, whereas its population status has never been assessed. We conducted, for the first time, field surveys for the adder at two areas of Southern Altay Mountains using a combination of line transects and random searches. We also described the morphological characteristics of the collected specimens and conducted analyses of external morphology and molecular phylogeny. The results showed that the adder distributed in both survey sites and we recorded a total of 34 sightings. In Kanas river valley, the estimated encounter rate over a total of 137 km transects was 0.15 ± 0.05 sightings/km. The occurrence of melanism was only 17%. The small size was typical for the adders in Southern Altay Mountains in contrast to other geographic populations of the nominate subspecies. A phylogenetic tree obtained by Bayesian Inference based on DNA sequences of the mitochondrial cytochrome *b* (1,023 bp) grouped them within the Northern clade of the species but failed to separate them from the subspecies *V. b. sachalinensis*. Our discovery extends the distribution range of *V. berus* and provides a basis for further researches. We discuss the hypothesis that the adder expands its distribution border to the southwest along the mountains' elevation gradient, but the population abundance declines gradually due to a drying climate.

## INTRODUCTION

As the most widely distributed terrestrial snake on the planet, the adder, *Vipera berus* Linnaeus, 1758, occupies nearly one-third of the Eurasian continent (*Saint Girons, 1980*; *Gasc et al., 1997*). Its range extends from Scotland (6°W) east to Sakhalin Island of the Russian Far East (143°E), and from Greece (42°N) north to northern Fennoscandia

(69°N) (*Nilson, Andrén & Szyndlar, 1994*; *Sillero et al., 2014*). Three subspecies are generally recognized: the nominate subspecies *V. b. berus* is found in most of the distribution range, *V. b. sachalinensis* is restricted to East Asia, and *V. b. bosniensis* occurs on the Balkan Peninsula, though some authors also consider *Vipera nikolskii* as a subspecies of the adder according to the biological species concept (*Milto & Zinenko, 2005*; *Zinenko, Ţurcanu & Strugariu, 2010*). The species inhabits diverse biotopes throughout northern temperate regions in various altitude ranges from sea level in the north up to the altitude of 2,600 m in the south (*Gasc et al., 1997*). Extensive studies on the adder have been published, for example on intraspecific taxonomy, mechanism of thermal melanism, and phylogeography (*Nilson, Andrén & Szyndlar, 1994*; *Ursenbacher et al., 2006*; *Clusella-Trullas, Van Wyk & Spotila, 2007*; *Joger et al., 2007*). However, most studies were concentrated on the European part of its distribution range; little is known from other regions, especially Central Asia. For example, in Mongolia, the adder has so far been found in fewer than 5 locations and the last study in northern Mongolia dates back 100 years ago (*Terbish et al., 2006*). A potential bias may exist in the ecological knowledge of the species, which might impede drawing generalizations, particularly when considering the geographical variation expressed throughout the entire range of the adder.

The southern boundary of taiga coniferous forests in Southern Altay Mountains is the southern distribution limit of the adder in Central Asia (*Bannikov et al., 1977*). Meanwhile, the easternmost distribution of steppe viper *V. renardi* extends to the northwest foothills located among Altay, Saur and Tarbagatai mountains (*Nilson & Andrén, 2001*; *Gvoždík et al., 2012*), where *Tuniyev, Nilson & Andrén (2010)* found a new species, *V. altaica*, which was closely related to but distinct from the parapatric *V. renardi* (see Fig. 1). The Mountains have also long been regarded as a transitional zone between the Siberian taiga forest and Central Asia steppe ecosystems, which are sensitive to climate change (*Xu et al., 2015*). The range limit of a species is its evolutionary response to the ecological constraints (*Holt & Keitt, 2005*). Therefore, studying the distribution of the adder in Southern Altay Mountains will help to understand the distribution patterns of reptiles and the influence of climate change on the boundary between the Euro-Siberian Subrealm and the Central Asia Subrealm (*Zhang, 1999*; *Ravkin, Bogomolova & Chesnokova, 2010*; *Zhou & Shi, 2015*). Comparative morphology and molecular phylogeny analyses for adders in Altay could shed light on its adaptation along the gradients of environment and the divergence and colonization history of the genus *Vipera*.

Recent phylogeographic studies have revealed that some widespread species are actually composed of multiple taxa, e.g., North American rat snake (*Burbrink, Lawson & Slowinski, 2000*), common kingsnake (*Alexander Pyron & Burbrink, 2009*), and steppe viper (*Ferchaud et al., 2012*). Although the published cladograms putatively represent all recognized taxa of the adder, it is possible that yet undiscovered taxa exist due to incomplete sampling (*Ursenbacher et al., 2006*). The taxonomy of adder population in Southern Altay Mountains is unclear and its geographic distribution remains unknown (*Bannikov et al., 1977*; *Zhao & Adler, 1993*; *Zhao, 2006*). So far, only three specimens (2 adults and 1 juvenile) have previously been recorded in China (*Zhao, Huang & Zong, 1998*). Indeed, the entire Altay

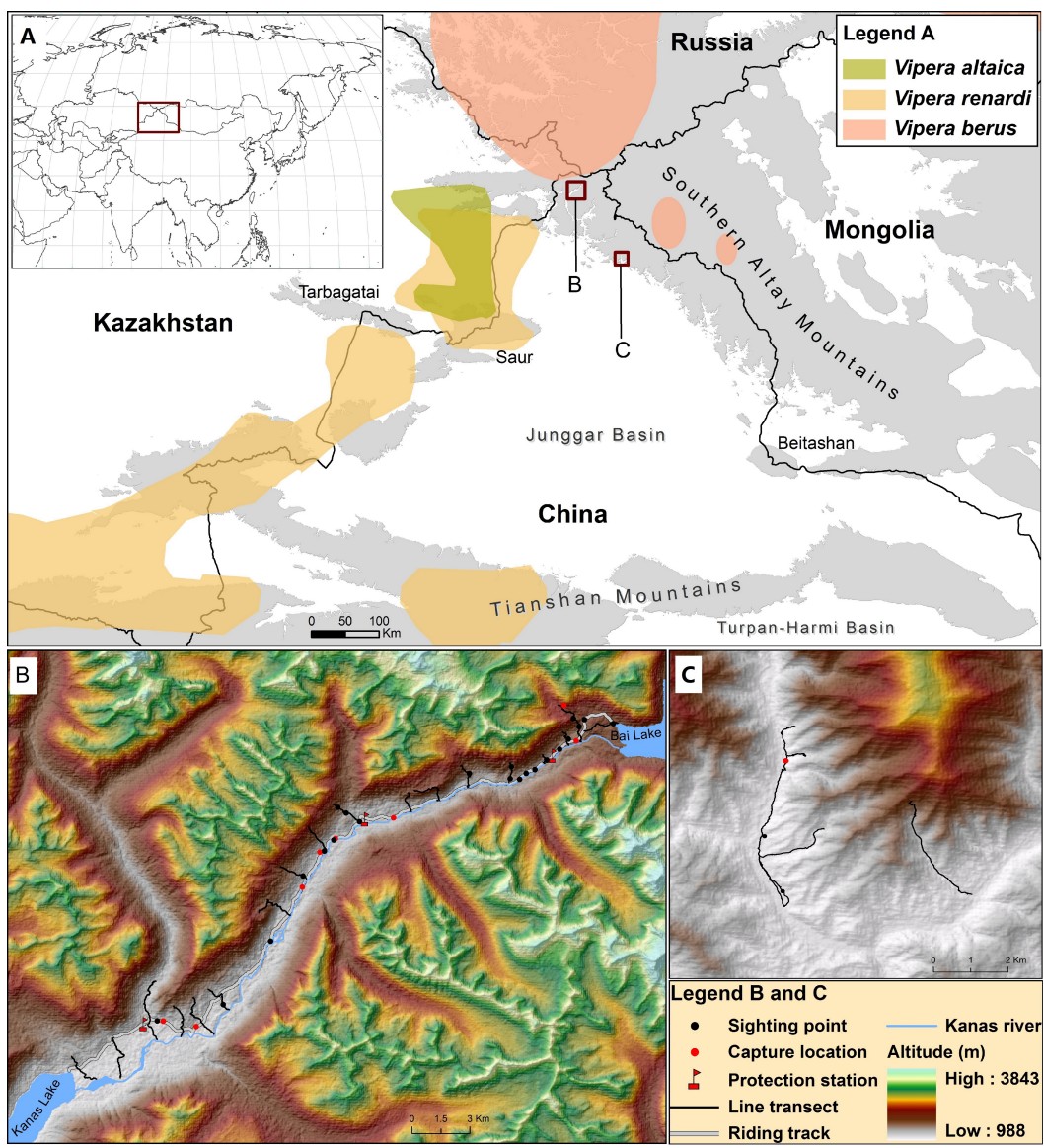

**Figure 1** **Map describing the distribution of *V. berus* and its relatives (*V. renardi* and *V. altaica*) in Southern Altay Mountains (A).** The detailed map of survey locations and field investigations is shown at Kanas river valley (B) and Daxiaodonggou Forested Area (C). The different distribution ranges are displayed following *Nilson, Andrén & Szyndlar (1994)*, *Nilson & Andrén (2001)*, *Terbish et al. (2006)*, *Ursenbacher et al. (2006)*, *Tuniyev, Nilson & Andrén (2010)* and *Gvoždík et al. (2012)*.

Mountains are regarded as the part of the distribution range of the adder but until now no detailed investigation has been conducted in the region.

In this study, we tried to obtain new data on geographic distribution, population characteristics, morphology and phylogenetic position of the poorly-known adder populations in Southern Altay Mountains. We collected specimens in the field and described the geographical variation in body dimensions, scalation and color pattern. We furthermore conducted phylogenetic analyses based on mitochondrial cytochrome *b* gene sequences to confirm the systematic position of the adder populations in the region.

## MATERIALS & METHODS

### Ethics statement

This study was reviewed and approved by the Ethical Committee of the Institute of Zoology, Chinese Academy of Sciences (Permit Number IOZ14060). All procedures performed in this study were in accordance with the instructions and permission of the Ethical Committee and the Chinese Wildlife Management Authority.

### Study area and field surveys

The study was conducted in the Southern Altay Mountains in the region bordering Russia, Mongolia and Kazakhstan. The elevation ranges from approximately 1,000 m in river valleys to 4,374 m at the summit. The study area has a typical temperate continental climate with short, cool summers and long, cold winters. The forest type is the boreal coniferous forests and the main tree species include Siberian spruce *Picea obovata*, Siberian larch *Larix sibirica*, Siberian pine *Pinus sibirica*, Siberian fir *Abies sibirica*, European aspen *Populus tremula*, and birch *Betula pendula*. The natural vegetation on the mountains varies with different altitude and aspect. The following vegetation belts can be recognized: coniferous forest, broad-leaved forest, mixed forest, shrubby steppe, subalpine shrub and meadow, alpine meadow, and nival belt.

We chose the Kanas river valley as the main survey area while we conducted a complementary research in Daxiaodonggou Forested Area (Fig. 1). The Kanas river valley in the northwest of Southern Altay Mountains is relatively close to the distribution range of the adder; the valley is strictly protected by three protection stations within the core area of Kanas National Nature Reserve. Daxiaodonggou Forested Area is located in the northwest-central part of the mountains, about 130 km from the main survey site. A combination of 20 line transects and random searches were conducted to investigate the abundance and distribution of adders in the two survey areas (Fig. 1). In Kanas river valley, we set up 6 line transects around each protection station (a total of 18 transects), which were spaced 1–2 km apart to assist spatial independence of snake observations. Transects began at the east of Kanas Lake and ended on the edge of Bai Lake. Each transect began where the forest or steppe edge met the river, aiming to representatively sample all habitats and exposures. The length and direction of transects were varied due to safety hazards from the terrain or vegetation (see Data S2). To investigate whether the adder was distributed in Daxiaodonggou Forested Area, we arbitrarily established 2 line transects to extend between the valley and hillside. Opportunistic sightings (e.g., travelling between protection stations along the riding track) were also recorded during fieldwork.

Except during rainy or foggy weather, we surveyed each transect at least twice between 10:00 and 19:00 h during a 170-day period (21 June–17 September 2014, 21 June–9 September 2015, see Data S2) by walking slowly (ca. 1–1.5 km/h). We searched mainly in open ground and under vegetation for adders. On all 18 line transects in Kanas river valley, we identified and recorded adders for the forward journeys only, to avoid counting the same individuals twice. At least 10 days were allowed between successive surveys on the same transect.

To standardize sampling, we used a simple encounter rate (number of sightings per km walked ± SE) as a proxy for relative species abundance. We could not distinguish black individuals between sexes based on the sexual dichromatism in the dorsal pattern due to melanism, so we did not determine the sex ratio of the targeted adder population in this study. We collected 9 specimens (3 males and 6 females) in Kanas river valley. We captured one adder from Daxiaodonggou Forested Area to determine the occurrence of the adder, and then released it at the point of capture. All the collected specimens were preserved in 95 or 100% ethanol and deposited in the herpetological collections of the Institute of Zoology, Chinese Academy of Sciences, Beijing, China. Collection information and voucher numbers are shown in Data S3.

## Morphological description and comparison

For the collected specimens, we described the morphological features in terms of body dimensions, scalation and color pattern. The measurements of eight characteristics traditionally used in *Vipera* systematics were recorded: apicals (refers to the apical scales, similarly hereinafter for the other different scales), supralabials, sublabials, circumoculars, crowns, loreals, ventrals, and subcaudals.

We also extracted the corresponding morphological data of related taxa from the available literature: *V. b. berus* from Northwestern, Southwestern and Eastern Europe (*Nilson, Andrén & Szyndlar, 1994*; *Milto & Zinenko, 2005*); *V. b. sachalinensis* from Northern Korea and Sakhalin (*Nilson, Andrén & Szyndlar, 1994*); *V. b. bosniensis* from Balkan (*Nilson, Andrén & Szyndlar, 1994*); *V. altaica* from Eastern Kazakhstan (*Tuniyev, Nilson & Andrén, 2010*); *V. renardi* from the eastern lowland of Central Asia (*Nilson & Andrén, 2001*; *Tuniyev, Nilson & Andrén, 2010*). Considering the sexual dimorphism present in pholidosis characteristics, both sexes were studied separately for ventrals and subcaudals.

We tested whether there were significant differences in the number of scales among different taxa from different geographical regions using a one-way analysis of variance (ANOVA), and we also performed a post-hoc Student–Newman–Keuls (SNK) multiple comparison tests to compare selected pairs of data. If data did not meet the criteria for homogeneity of variance, two-tailed unequal variance $t$-test followed by Bonferroni correction was performed (*Ruxton, 2006*). All statistical approaches were completed using R v3.1.2 (*R Core Team, 2013*). $P < 0.05$ was regarded as statistically significant. However, because of the small sample size, we did not conduct statistical analyses on the ventrals and subcaudals from different sexes.

## Molecular phylogenetic analyses

Genomic DNA was extracted from tissue samples of each specimen using TIANamp Genomic DNA Kit (Tiangen Biotech, Beijing, China), following the manufacturer's instructions. We amplified and sequenced partial mitochondrial cytochrome $b$ (Cyt $b$) gene with the primers (*Ursenbacher et al., 2006*):

L14724Vb (5′-GATCTGAAAAACCACCGTTG-3′),
H15548Vb (5′-AATAGAAAGTATCATTCTGGTTTAAT-3′),
L15162Vb (5′-CTCCCATGAGGACAAATATC-3′),
and H15914Vb (5′-CCAGCTTTGGTTTACAAGAAC-3′).

Polymerase chain reaction (PCR) was performed in a 40 µL final volume with 4 µL DNA template, 20 µL of Ex Taq polymerase (RR001A, TaKaRa), 4 µL (20 mmol/L) of each of the two primers and 8 µL of double distilled water. PCR programs to amplify the two parts of Cyt *b* started with a cycle of denaturing at 94 °C for 10 min, followed by 35 cycle of 94 °C for 45 s, 50 °C for 60 s, and 72 °C for 90 s, and a final extension at 72 °C for 10 min. PCR products were sequenced by Sino Geno Max (Beijing, China), using ABI 3730XL sequencers. Sequences were deposited in GenBank (accession numbers KU942378, KX345249, KX345250, KX345251, KX345252, KX345253, KX345254, KX345255 and KX345256).

Cyt *b* sequences were visualized with MEGA 6.0 (*Tamura et al., 2013*), aligned with the ClustalW option included in this software and double-checked by eyes (*Larkin et al., 2007*). The number of haplotypes and polymorphic sites was obtained with DnaSP v5.0 (*Librado & Rozas, 2009*). Only one haplotype was recognized in all 9 samples and compared to other 33 sequences downloaded from GenBank. We used *V. seoanei*, *V. altaica*, and *V. renardi* as outgroups (see Table S1). Sequence differences were calculated using MEGA 6.0 based on Kimura's two-parameter model (*Tamura et al., 2013*). The best-fit model of nucleotide substitution was calculated by the Akaike Information Criterion (AIC) approach in Modeltest v3.7 (*Posada & Crandall, 1998*). Then, we conducted the Bayesian inference (BI) of phylogeny with MrBayes v3.2 (*Ronquist & Huelsenbeck, 2003*) using the best-fitted model. Four chains were run simultaneously and each Markov chain was started from a random tree and run for $1 \times 10^7$ cycles, with sampling every 1,000th cycle. The convergence was estimated by measuring the standard deviation of the split frequency among parallel chains. Chains were considered to have converged when the average split frequency was lower than 0.01. The initial 25% trees were discarded as burn-in values.

## RESULTS

### Field surveys

In the two survey sites, we recorded a total of 34 sightings of adders, of which 31 were from Kanas river valley (17 and 14 in 2014 and 2015, respectively) and 3 from Daxiaodonggou Forested Area. There were 29 sightings of subadults and adults (henceforth adults), 1 juvenile trampled by a horse with body length of 270 mm, 1 newborn in Kanas valley, and 3 adults in Daxiaodonggou Forested Area. However, we did not mark the encountered adders in the field survey, so the same individual was likely counted multiple times. In Kanas river valley, a total of 105 km of line transects was surveyed and the overall encounter rate was 0.15 ± 0.05 sightings/km. Melanistic adders were relatively rare with only 17% ($n = 5$) of the total number of adults. The adders were found mostly in coniferous forest (18 out of 34 sightings, 53%) at elevations ranging from 1,200 m to 2,300 m (see Data S1). There was only one snake species (i.e., *V. berus*) observed in Kanas valley which is about 80 km from the nearest population of *V. renardi* (Fig. 1), but *Elaphe dione* and *Coluber spinalis* occurred in Daxiaodonggou Forested Area.

## Morphological characteristics

The largest specimen (voucher number IOZ015018) examined by us was a female measuring 510 mm in total length and 54 mm from the posterior margin of cloacal opening to the tip of tail (tail length); the latter was equal to 10.6% of body length. The largest male (IOZ015020) had a total length of 490 mm, including 68 mm tail length (13.9% of total length). The males averaged at a total length of $449.3 \pm 21.5$ mm ($n = 3$) while the mean length of the females was $487.8 \pm 6.7$ mm ($n = 5$). Some morphological characteristics were identical in all collected specimens: 2 large supraoculars and 1 large frontal plate on top of head, parietals divided, 1 canthal and 1 supranasal scale on each canthus rostralis, 1 scale between nasal and rostral on each side, 2 apicals in contact with rostral, and a single row of scales between supralabials and eye. However, individual variation was evident in some scalation characters as follows: 4 to 10 intercanthals, 5 to 14 intersupraoculars, 2 to 5 loreals on each side, normally 9 to 10 supralabials and 9 to 11 sublabials on each side, and 1 to 3 preventrals. The eyes were surrounded by 18 to 25 circumoculars; the rostral bordered by 2 supralabials. 144 to 150 ventrals and 31 to 38 subcaudals in females, 140 to 146 ventrals and 37 to 40 subcaudals in males, all subcaudals in double rows, 20 to 21 rows of dorsal scales around midbody. Detailed morphological data of each specimen with the corresponding voucher number can be found in Data S3.

Dorsal pattern consisted of a continuous winding zig-zag band in females and males. Lateral body blotches were present except in melanistic males, ventral side black, the tip of tail pigmented by yellow or orange. Preoculars and frontal were separated from nasal and supraoculars on each side, respectively. Females, whether or not they were melanistic, had light dots on ventrals and reddish-brown colored throat. Non-melanistic specimens had two dark oblique bands on the dorsal head and light spots on some supralabials, apicals, supranasals, and supraoculars.

Sexual dichromatism in the dorsal pattern was well expressed. All non-melanistic females had a dark brown zig-zag band on light brown background. Compared with that, males showed a much higher color contrast in dorsal pigmentation, usually with black zig-zag turns and greyish ground color.

## Morphological comparison

The mean values and standard errors of morphological characteristics, which were calculated for 9 specimens in this study and extracted from literature for the samples from other regions, are shown in Fig. 2 and Table 1. The specimens examined by us were distinguishable from *V. altaica* and *V. renardi* in the character of apicals, crowns, and loreals (all $P < 0.05$). At the subspecies level, the results demonstrated geographical variation in morphological characteristics. Taken as a whole, the adders from Southern Altay Mountains exhibited a tendency toward increased number of apicals. They significantly differed from the *sachalinensis* subspecies in apicals, sublabials and crowns, and *bosniensis* in sublabials and loreals (all $P < 0.05$). When compared with other specific populations of the nominate subspecies from different regions, our materials only showed a difference with Northwestern Europe and Southwestern Europe populations in supralabials and crowns, respectively; they had no difference with adders from Eastern Europe. In addition, the

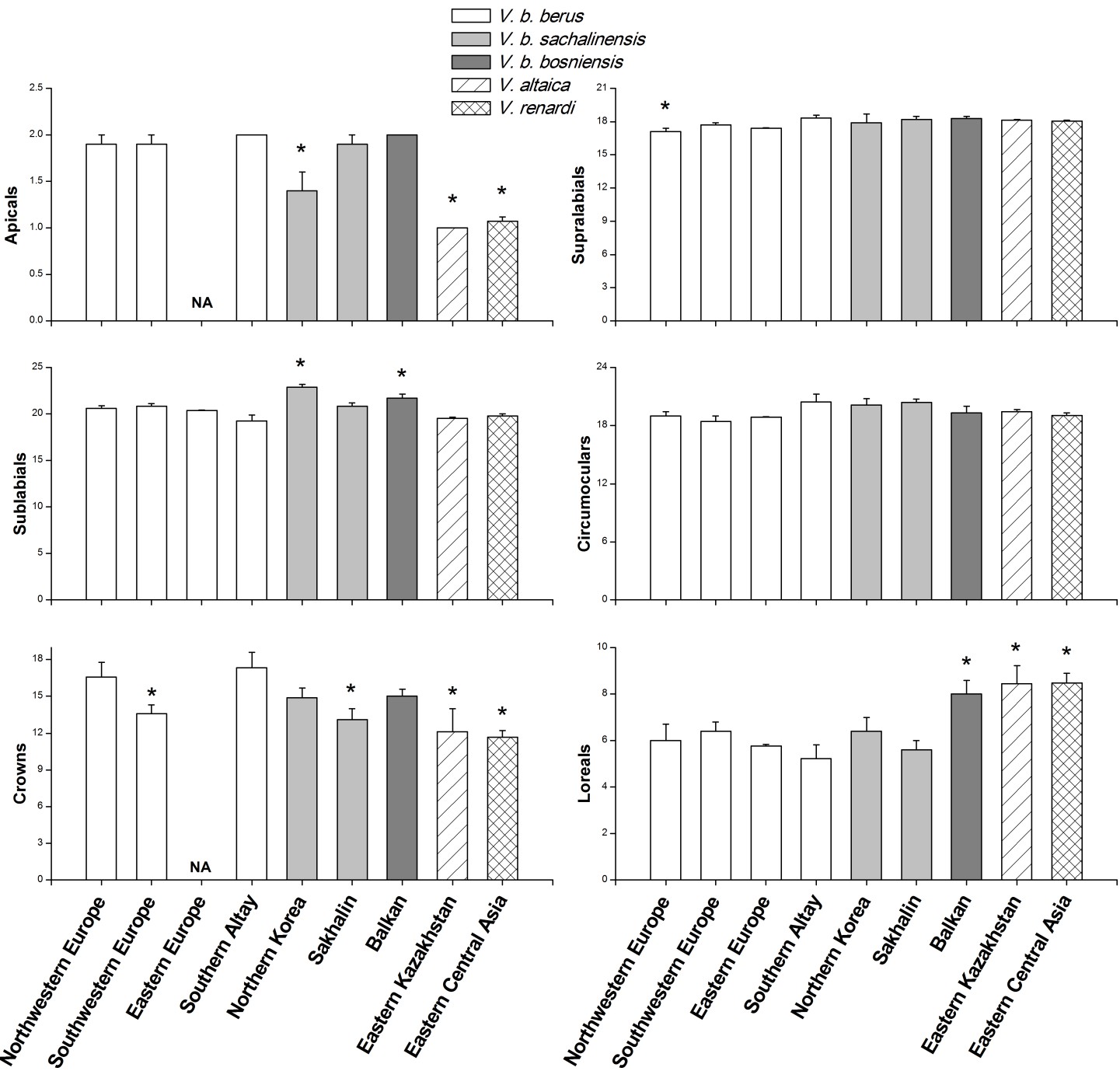

**Figure 2** **Comparisons of morphological characteristics of *V. berus* and related species.** Asterisk (*) represents a significant difference between our specimens from Southern Altay Mountains and other taxa from different geographical regions (multiple comparison tests, $P < 0.05$). Data on the number of apicals and crowns in the adder population from Eastern Europe were not available (NA). The source and sample sizes ($n$) of the morphological data of different taxa are as follows: *V. b. berus* from Northwestern and Southwestern Europe ($n = 19$ and $14$, respectively; *Nilson, Andrén & Szyndlar, 1994*), Eastern Europe ($n = 45$; *Milto & Zinenko, 2005*), and Southern Altay ($n = 9$; this study, voucher numbers listed in Data S3); *V. b. sachalinensis* from Northern Korea and Sakhalin ($n = 12$ and $10$, respectively; *Nilson, Andrén & Szyndlar, 1994*); *V. b. bosniensis* from Balkan ($n = 15$; *Nilson, Andrén & Szyndlar, 1994*); *V. altaica* from Eastern Kazakhstan ($n = 38$; *Tuniyev, Nilson & Andrén, 2010*); *V. renardi* from Eastern Central Asia ($n = 30$; *Nilson & Andrén, 2001*; *Tuniyev, Nilson & Andrén, 2010*).

**Table 1  Comparisons of ventrals and subcaudals of *V. berus* from different geographical regions.**  The source of the morphological data of different taxa is shown in Fig. 2.

| Taxon | Location | Males | | | Females | | |
|-------|----------|-------|---|---|---------|---|---|
| | | N | Ventrals | Subcaudals | N | Ventrals | Subcaudals |
| *V. b. berus* | Northwestern Europe | 10 | 140.7 ± 0.9 | 38.0 ± 0.6 | 4 | 145.5 ± 3.2 | 30.3 ± 0.8 |
| | Southwestern Europe | 11 | 139.5 ± 1.2 | 34.6 ± 1.4 | 3 | 141.3 ± 1.5 | 25.3 ± 1.5 |
| | Eastern Europe | 22 | 144.5 ± 0.4 | 38.9 ± 0.5 | 23 | 148.6 ± 0.6 | 31.1 ± 0.4 |
| | **Southern Altay** | **3** | **143.0 ± 1.7** | **38.7 ± 0.9** | **6** | **146.2 ± 1.1** | **33.7 ± 1.1** |
| *V. b. sachalinensis* | Northern Korea | 4 | 144.7 ± 1.3 | 35.2 ± 1.1 | 6 | 151.8 ± 0.9 | 31.2 ± 0.9 |
| | Sakhalin | 7 | 146.9 ± 1.1 | 37.3 ± 0.9 | 3 | 154.7 ± 0.9 | 28.3 ± 0.7 |
| *V. b. bosniensis* | Balkan | 9 | 142.1 ± 0.9 | 37.1 ± 1.4 | 6 | 143.7 ± 1.9 | 30.2 ± 0.7 |
| *V. altaica* | Eastern Kazakhstan | 16 | | 35.4 ± 0.3 | 21 | | 27.2 ± 0.3 |

**Notes.**
The collected specimens in this study are shown in bold.

maximal number of crown shields of head and minimal number of loreals and sublabials were recorded in our specimens.

Males and females expressed sexual dimorphism in the number of ventrals and subcaudals (Table 1). The adder from Southern Altay Mountains was similar to *V. b. berus* populations from Northwestern Europe and Eastern Europe and *V. b. bosniensis* population from Balkan in both female and male values; they showed much increased number of subcaudals compared with the nominate subspecies in Southwestern Europe and *sachalinensis* subspecies. Our samples also differed from *V. b. sachalinensis* in female's ventrals. *V. altaica* resembled our specimens in ventrals (the average number is 145.9 in *V. altaica*, *Tuniyev, Nilson & Andrén (2010)* pooled both sexes when analyzing morphological data), but it had a much lower average number of subcaudals, especially in females.

## Molecular phylogenetic analyses

The alignment of the mtDNA Cyt *b* (1,023 bp) of the adders contained 119 variable sites and 61 parsimony informative sites. All samples in our study shared the same haplotype, which was identical to that from Perm in Russia (see Data S2).

The Bayesian phylogenetic tree clearly separated three major lineages of the adder, the Italian clade, Balkan clade and Northern clade, each receiving high statistical support (Fig. 3). The three currently accepted subspecies did not represent distinct genetic lineages; only the Balkan clade conformed to the *bosniensis* subspecies. The majority of haplotypes of *V. b. berus* grouped within the Northern clade while the haplotypes of the populations from Italy and adjacent areas were clustered together within the Italian clade. The adder from Southern Altay Mountains was grouped into the Northern clade and clearly separated from the other two clades. Moreover, the BI reconstruction was not able to separate the *sachalinensis* subspecies from the nominate subspecies.

The Kimura two-parameter distances among all *Vipera berus* were displayed in Table 2. The genetic distance between the Southern Altay group and the Northern clade was 0.003 ± 0.001, which was lower than any other pairwise distances. However, a low degree of

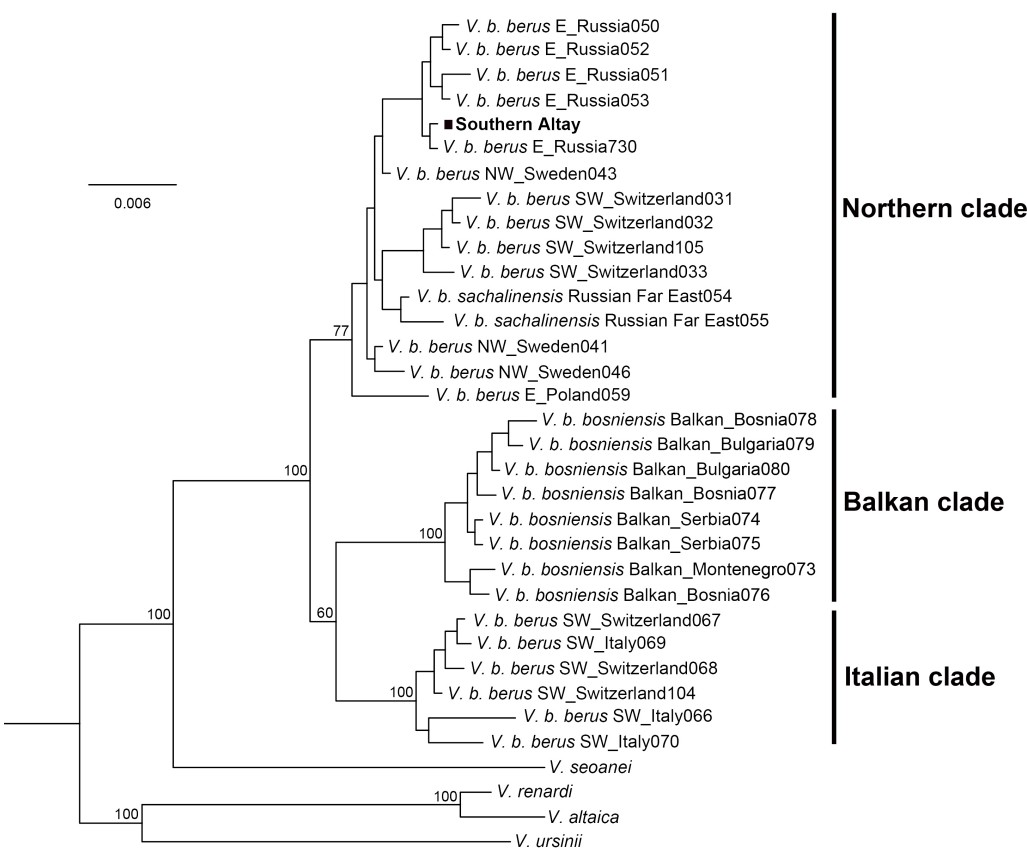

**Figure 3 Phylogenetic tree of cytochrome *b* obtained from Bayesian analysis.** Bayesian posterior probability is shown for the main nodes. The geographic origin of each haplotype is indicated (E, Eastern Europe; SW, Southwestern Europe; NW, Northwestern Europe). The samples from Southern Altay Mountains are highlighted by the black block. See Table S1 for sample codes, locations and accession numbers.

**Table 2 Genetic distance of *Vipera berus* among different geographic populations.**

|  | Southern Altay | Northern clade | Far East | Italian clade |
|---|---|---|---|---|
| Northern clade | $0.003 \pm 0.001$ |  |  |  |
| Far East | $0.005 \pm 0.002$ | $0.005 \pm 0.002$ |  |  |
| Italian clade | $0.014 \pm 0.004$ | $0.014 \pm 0.003$ | $0.014 \pm 0.004$ |  |
| Balkan clade | $0.014 \pm 0.004$ | $0.014 \pm 0.004$ | $0.014 \pm 0.004$ | $0.017 \pm 0.004$ |

**Notes.**

Sequences are grouped together based on the phylogenetic results but samples of Southern Altay Mountains and Russia Far East are treated as two separated groups.

genetic differentiation ($0.005 \pm 0.002$) was also observed between our specimens and the *sachalinensis* subspecies.

## DISCUSSION

The adders observed during line transect surveys represent an estimated encounter rate of 0.15 sightings per km in Southern Altay Mountains. The relatively low number of sightings

($n = 13$) may reduce the reliability of abundance estimates since the minimum sample size of 40 (*Buckland et al., 2001*) has not been reached. However, such a number may be difficult to achieve due to the scarcity of adders (*Terbish, Munkhbayar & Munkhbaatar, 2006*), even if intensive surveys are considered (in this study the riding track was surveyed many times during fieldwork). Unfortunately, as there is no previous report on the abundance of adders in Southern Altay Mountains (there are only a few reports of its occurrence, *Zhao & Adler, 1993*; *Zhao, Huang & Zong, 1998*), we cannot compare our results with others.

The occurrence of melanism is low in the studied adder population, among which only 17% of all observed adults are melanistic. This ratio differs from most results obtained from other geographic populations in previous studies (e.g., 47% in southwest Sweden, *Andrén & Nilson, 1981*; 49% in the Carnic Alps, *Luiselli, 1992*; 47% in the Swiss Alps, *Monney, Luiselli & Capula, 1995*; but 16% in the Northern Romanian Carpathians, *Strugariu & Zamfirescu, 2011*). The color of ectothermic vertebrates plays a significant role in thermoregulation processes (*Clusella-Trullas, Van Wyk & Spotila, 2007*). The thermal melanism hypothesis posits that melanistic individuals have benefits in cool habitats because, with low skin reflectance, they heat up faster and reach higher equilibrium temperatures than normal individuals (*Clusella-Trullas, Van Wyk & Spotila, 2007*). This hypothesis has been widely studied in ectothermic species, especially the color-polymorphic adder (*Andrén & Nilson, 1981*) and *Vipera aspis* (*Castella et al., 2013*). Thus, considering the southernmost marginal location of our study area in the adder's distribution range of Central Asia, the observed low occurrence of melanism may be a result of a much warmer climate. However, as *Strugariu & Zamfirescu (2011)* argued, such a speculation may only provide a partial explanation, because the Kanas river valley has a typical continental climate with an average summer temperature of 15.0 °C, slightly colder than both the Northern Romanian Carpathians and southwest Sweden (summer average temperature of 15.4 °C and 16.1 °C, respectively; *Hijmans et al., 2005*). *Andrén & Nilson (1981)* argued that the presence of melanism might depend on the trade-off between thermoregulatory benefit and predation cost.

The adder exhibits a low level of geographically correlated morphological variation, especially the nominate subspecies with a rather homogeneous scalation characteristics (Fig. 2). However, the mean length of the adult adder (males and females averaged at 44.93 cm and 48.78 cm respectively) in Southern Altay Mountains is smallest among those presented by other similar studies (around 56 cm and 62 cm in the eastern Swedish islands, *Forsman & Ås, 1987*; 55 cm and 65 cm in northern England, *Phelps, 2004*; 55 cm and 62 cm in southwestern Sweden, *Andrén & Nilson, 1981*; 47 cm and 52 cm in the Swiss Alps, *Monney, Luiselli & Capula, 1995*; 44.98 cm and 51.87 cm in the northern Romanian Carpathians, *Strugariu & Zamfirescu, 2011*). This relatively atypical size, together with the higher number of crowns and subcaudals (only in females) and lower number of loreals and sublabials, could be considered autapomorphic traits of the adder on Southern Altay Mountains at the southern distribution margin. It is generally accepted that the geographical variation of external traits results from historical (i.e., phylogenetic) and environmental (i.e., adaptive) factors, first by population isolation in geographic refuges and further by local adaptation to divergent environments (*Thorpe et al., 1991*; *Malhotra & Thorpe, 1997*; *Santos et al., 2014*). For example, many studies have demonstrated that
the number of ventral scales in snakes is strongly related to the geographic origin (*Dohm & Garland, 1993*; *Fornasiero et al., 2007*), but also that the thermal condition can influence the within-population ventral scale variation during embryogenesis (*Osgood, 1978*; *Lourdais et al., 2004*).

The detailed morphological analysis for determining the evolutionary history of a group is preferred over the labile characteristics, as color patterns do in the North American rat snake (*Burbrink, Lawson & Slowinski, 2000*). *Tóth & Farkas (2004)* determined three typical features for the subspecies *bosniensis*, including two subocular scale rows, high percent of specimens with interrupted zig-zag pattern, and a higher row number of dorsal scales around midbody. For the subspecies *sachalinensis*, two characteristics were previously proposed as diagnostic by *Nilson, Andrén & Szyndlar (1994)*: frontal in contact with supraoculars and upper preocular in contact with nasal. Our specimens did not show these special characteristics of the two subspecies. Therefore, although the applicability of these characteristics for determination is still questioned in certain regions (e.g., *Zhao, Huang & Zong, 1998*; *Zhao, 2006*), they can be called "working" in Southern Altay Mountains. This result could allow an expected conclusion that, on a morphological basis, the adder in Southern Altay Mountains differs from *V. b. bosniensis* and *V. b. sachalinensis* and belongs to *V. b. berus*. It is well accorded with the real situation that the survey area of Kanas river valley is located near the distribution range of the nominate subspecies in the border region of Russia and Kazakhstan, and also covered by a same Siberian taiga habitat (*Zhang, 1999*; *Ravkin, Bogomolova & Chesnokova, 2010*).

The adder population from Southern Altay Mountains was grouped into the Northern clade together with those populations from Eastern Europe to Sakhalin Island of the Russian Far East across the adders' range (Fig. 3). Despite the southernmost marginal distribution in Central Asia, phylogenetically the population was not classified into a separate group, supporting the presented phylogenetic relationships (*Ursenbacher et al., 2006*). Our samples were clearly separated from the Balkan clade and Italian clade, in line with the morphological results. The few morphological similarities between them (e.g., in supralabials) may be caused by either parallel evolution (parallelism) or ancestral characteristics (symplesiomorphy) (*Nilson, Andrén & Szyndlar, 1994*; *Ursenbacher et al., 2006*). The low degree of genetic differentiation between adders from Southern Altay Mountains and the *sachalinensis* subspecies indicates complex recolonization dynamics of the Eurasia continent by *V. berus* (*Ursenbacher et al., 2006*). The mitochondrial DNA alone may be less informative for distinguishing the taxon *sachalinensis* in DNA sequence–based methods (*Ursenbacher et al., 2006*; *Wares, 2014*). The origin and splitting history of *V. sachalinensis* are yet poorly understood, neither does the subspecific boundary (*Bannikov et al., 1977*), partly due to scanty field work towards this subspecies.

*V. altaica* lives in lowland and foothill habitats in the altitude range of 200–1,200 m, but is characterized by having the morphological characteristics typical for montane taxa. Its range occurs at the parapatric boundary of *V. renardi* and *V. berus* (*Tuniyev, Nilson & Andrén, 2010*). One possible condition is that this doubtful species is an assemblage of hybrids between *V. renardi* and *V. berus*. However, we did not find *V. altaica* expressed intermediate morphologies and its special features compared with eastern lowland *V. renardi*, including

smallest size (maximum total length is 38.8 cm for males, and 39.9 cm for females), non-bilineate dorsal pattern, white belly and no suture on the supralabials, also differed from those of our specimens. Furthermore, the number of apicals is a key discriminator as the adder has more apicals than the steppe viper, while all *altaica* samples have only one apical (*Tuniyev, Nilson & Andrén, 2010*). Also, there are no melanistic individuals observed in the taxon *altaica*. Phylogenetic analyses revealed that *V. altaica* was grouped with *V. renardi*, and no mtDNA introgression detected in this study. To confirm the hybrid hypothesis, more evidence (e.g., an analysis of nuclear gene data) is required. Therefore, as has been previously suggested, *V. altaica* may be a relict species (*Tuniyev, Nilson & Andrén, 2010*; *Zinenko et al., 2015*).

The early accounts of *Zhao, Huang & Zong (1998)* reported finds of the adder in Fuhai County about 70 kilometers southeast of Daxiaodonggou Forested Area, indicating that the adder expands its distribution from northwest to southeast along the elevation gradient in the Southern Altay Mountains. However, the eastern mountains of the Junggar basin create a barrier effect that restricts the eastward penetration of Atlantic westerlies (Fig. 1). This factor, together with the influence of arid climate in Mongolia, leads to an increasing continentality and decreasing precipitation (*Wei et al., 2003*). The drying climate changes the structure and composition of the vegetation, so it is reasonable to suppose that the population abundance of adders declines gradually towards the southeast along the Mountains The dominant role of the adder may be replaced by other snake species (e.g., *Elaphe dione* and *Coluber spinalis*), as described in this study. However, the existing data concerning the adder over the Southern Altay Mountains was too scarce to corroborate the hypothesis. Further research can explore the geographic range limit of the adder by investigating its population dynamics from Daxiaodonggou Forested Area southeastward along the mountains.

Despite occupying a wide distribution, the adder is often found restricted to highly fragmented populations and isolated mountainous relicts (*Schiemenz, 1995*; *Sillero et al., 2014*). Human activities have resulted in a massive reduction in population size in certain regions, such as Smygehuk in southern Sweden where, due to agricultural activities, an adder population is clearly separated from neighboring populations varying in size from less than 50 to more than 250 adult individuals (*Madsen, Stille & Shine, 1996* and references therein). Our study highlights a fundamental issue regarding the assessment of the conservation status of the poorly-known species in Central Asia. The adder is also poorly studied but believed to be experiencing a decline in habitat quality and extent in Mongolia (*Terbish et al., 2006*). The last study in northern Mongolia dates from 100 years ago. *Terbish et al. (2006)* categorized the adder as Vulnerable according to IUCN Red List criteria at regional levels, because it is estimated to be found in fewer than 5 locations within an estimated area of 6,000 km$^2$. Only approximately 9% of the species' range in Mongolia lies within protected areas, mostly in Mongolian Altay Mountains (i.e., the opposite (north) slope of our study areas) (*Terbish, Munkhbayar & Munkhbaatar, 2006*). Little is known about the threats to the adder in both China and Mongolia. Clearly, extensive research is recommended to determine the status and threats of the species, and further to develop management plans in these areas.

## ACKNOWLEDGEMENTS

We are grateful to Zhang Liang for helpful advice on the manuscript. We thank Li Lili, Qi Yingjie, Li Yubing, Li Na, Ji Shengnan, Tao Xiaqiu, and staff from Kanas Nature Reserve for assistance during fieldwork. Thanks also to Ulrich Joger and an anonymous reviewer for insightful comments that improved the manuscript.

### Funding

This study was supported by the Basic Science Special Project of Ministry of Science and Technology of China (2013FY110300), the Knowledge Innovation Project of the Chinese Academy of Sciences (KSCX2-EW-J-2), and the National Nature Science Foundation of China (31572260). The funders had no role in study design, data collection and analysis, decision to publish, or preparation of the manuscript.

### Grant Disclosures

The following grant information was disclosed by the authors:
Basic Science Special Project of Ministry of Science and Technology of China: 2013FY110300.
Knowledge Innovation Project of the Chinese Academy of Sciences: KSCX2-EW-J-2.
National Nature Science Foundation of China: 31572260.

### Competing Interests

The authors declare there are no competing interests.

### Author Contributions

- Shaopeng Cui conceived and designed the experiments, performed the experiments, analyzed the data, contributed reagents/materials/analysis tools, wrote the paper, prepared figures and/or tables, reviewed drafts of the paper.
- Xiao Luo performed the experiments, analyzed the data, contributed reagents/materials/analysis tools, wrote the paper, prepared figures and/or tables.
- Daiqiang Chen conceived and designed the experiments, performed the experiments.
- Jizhou Sun performed the experiments, contributed reagents/materials/analysis tools, reviewed drafts of the paper.
- Hongjun Chu contributed reagents/materials/analysis tools, reviewed drafts of the paper.
- Chunwang Li and Zhigang Jiang conceived and designed the experiments, reviewed drafts of the paper.

### Animal Ethics

The following information was supplied relating to ethical approvals (i.e., approving body and any reference numbers):

This study was reviewed and approved by the Ethical Committee of the Institute of Zoology, Chinese Academy of Sciences (Permit Number IOZ14060). All procedures

performed in this study were in accordance with the instructions and permission of the Ethical Committee and the Chinese Wildlife Management Authority.

## DNA Deposition

The following information was supplied regarding the deposition of DNA sequences:

The sequence was deposited in GenBank with Accession No. KU942378, KX345249, KX345250, KX345251, KX345252, KX345253, KX345254, KX345255 and KX345256. See Supplemental Information.

## Data Availability

The raw data has been supplied as a Supplemental Dataset.

## Supplemental Information

Supplemental information for this article can be found online at http://dx.doi.org/10.7717/peerj.2342#supplemental-information.

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
