# Peer review of "The adder (Vipera berus) in Southern Altay Mountains: population characteristics, distribution, morphology and phylogenetic position"

_PeerJ, doi:10.7717/peerj.2342_

## Round 0.1 · original submission · Major Revisions

· Academic Editor

Major Revisions

Dear authors

Thank you for submitting your manuscript to our journal. As you see our reviewers suggest a major revision of your ms. If you are willing to do so, please respond to their points and we would be happy to reconsider your revised manuscript.

Michael Wink
Academic editor

·

Basic reporting

This is a short report on Vipera berus poupaltions in the Chinese part of the Altai mts. There is a phylogenetic tree contructed with mitochondrial Cytb sequences from Genebank with own material included, as well as morphological data presented.

Experimental design

The fieldwork was rather simply designed. Individuas were not marked or photographed in order to avoid multiple recording of the same specimen. So no calcuklatiion of population size was possible. The data on abundance are therefore of limited value only.

The morphologicalcomparison was done with other Vipera berus only. It would have been better to include Vipera ataica, as this doubtful species occurs in the same region, and it shouws an intermediate morrphology between V. berus and V. renardi.

The molecular design was ok, as far as I can judge. Vipera altaica sequences have not yet been published.

Validity of the findings

The findings are of limited value (see above). Therefore I suggest to include V. altaica in the morphological study. This could lead to different results. Especially, the open question whether V. altaica is a species or an assemblage of hybrids between V. renardi and V. berus needs to be adressed.
The manuscript should be re-submitted after altaica was included in table 2 and disdcussed properly.

Other points see below.

Additional comments

A few corrections, comments and questions:
Line 135: What means '*fine days". Do you went out only in good weather?
Line 142: "We could not distinguish between sexes": This refers only to black specimens, as you differentiated sex in others (as in line 161).
Line 197: 'Strampled" means "trampled", I guess.
Line 203: Please mention how far the nearest population of Vipera renardi is from your sites.
Line 255: Your report about the haplotypes ofItaly is not correct. Some appear in the northern clade, others not. They should not be in the northern clade at all. This must be discussed.
Line 291: Average temperature is not relevant for a snake that hibernates for half of the year. Mean summer temperature or No. of sunny days should be taken instead.
Line 293: Anren & Nilson had published this trade-off much earlier than Broennimann et al.
Line 346: 'classified into a single group' - you mean a separate group, don't you?
Line 355: 'distingish the schalinensis taxa: correct would be 'distinguishing the taxon sachalinensis'.
Line 357: subspecific boundary
Line 366/367: Drying of the clinmate does not mean that the adder expands. Instead, V. renardi would expand on the expense of V. berus.

Reviewer 2 ·

Basic reporting

This manuscript reports data on Vipera berus from its eastern distribution range. Given the scarcity of data from this region, these data clearly are worth reporting. The manuscript presents a variety of data, including the results of transect counts in the field, morphological information, and mitochondrial DNA information.

Although the information presented is of interest and worth publishing, I found the manuscript a bit wordy considering the few data that are here presented. One could also argue that this information could be instead published in a one-page short note in a journal such as Herpetological Review or Herpetology Notes. To make the manuscript publishable in PeerJ, I would recommend at least strongly condensing the Introduction and the Discussion, and not trying in these two sections to claim more importance to the study than it actually has. Many general statements can just be removed.

I have made a large number of corrections directly in the PDF and I attach the edited version. Please revise the text accordingly. However, I have almost not touched the Discussion, simply because I think it first needs to be strongly condensed, at least by 30%, better to half of its current length.

Some additional concerns and suggestions, to be considered during the preparation of a revised manuscript:

Voucher specimens: In which collection have the collected specimens been deposited? It would be of great importance to deposit these specimens in a publically available collection, such as a natural history museum, and report here the voucher specimen numbers.

Genbank accession number: This I don't understand. You say you sequenced nine individuals, but you only submitted one sequence to Genbank. Even if all sequences were identical, I would like to request to submit all nine sequences to Genbank with their respective voucher numbers and precise locality information. This will make it easier for subsequent researchers using these sequences to evaluate the amount of genetic variation (in this case zero) in different populations, without having to look up all of this the information in every publication.

One thing I don't understand is how you can be sure that you saw 34 adders. Is this 34 sightings (with some specimens maybe sighted multiple times? Can you exclude that you have seen the same adder multiple times? Have you marked them to see if they were recaptures? If not, it will be necessary that you very clearly speak about adder sightings rather than adders, and include a statement that you cannot exclude having scored the same specimen multiple times.

The morphological comparisons to other subspecies are fine and important, but I am surprised the authors do not even try to apply any statistical procedure here. This is something I clearly would request. Probably the most obvious would be a Principal Component Analysis of all the scalation characters, and then providing a scatterplot in which the specimens of each of the comparative populations is shown in a different color... this will easily allow assessing whether the Altay population is morphologically differentiated, how well V. berus populations are in general differentiated, and which is the most similar population to the Altay population.

Tables with morphological characters: Please make sure to present in a crystal clear way the source of every data you are presenting here. If these are specimens you have measured/counted, then make sure to present a full list of specimens examined along with their voucher specimen / collection numbers.

Experimental design

See above

Validity of the findings

See above

Additional comments

See above

Annotated reviews are not available for download in order to protect the identity of reviewers who chose to remain anonymous.

---

## Round 0.2 · Minor Revisions

· Academic Editor

Minor Revisions

Dear authors

One of our reviewers would like to see a few more changes.
We look forward to obtain your revision

Regards

Michael Wink
Academic editor

·

Basic reporting

The revised manuscript considers all the points I criticized before. It is now ready for being published.

Experimental design

ok

Validity of the findings

ok

Additional comments

Thank you for doing such a good work.

Reviewer 2 ·

Basic reporting

The manuscript has been much improved during the revision, and the authors did a very good job in explaining the changes applied – thank you for that.

I have only a few additional suggestions that the authors can be trusted to apply in the final manuscript. I strongly recommend, in addition, that the authors take out a final intensive proofreading of the manuscript since it still contains a number of minor problems in grammar and style.

Figure 2: Could you please place over the bars the sample size studied for each of the populations/subspecies (or mention the sample sizes in the figure caption).

Line 185, please report here in addition the range of accession numbers
(such as: ######-######). This way authors wanting to check the sequences do not need to refer to supplementary documents.
And: please change sentence to read: Sequences were deposited ...

Line 217-220. Please mention collection numbers of specimens here.

Line 221. In the rebuttal letter you say, all meristic characters refer to a single specimen. Please give here the voucher number of the examined specimen to whcih the subsequent scale counts refer.

Line 244: are shown

Line 265: observed --> contained

Line 267-268, remove mention of Genbank accession number here)

Line 272, change „V. b. berus's haplotypes” to “haplotypes of V. b. berus”

Line 278, the Southern Altay group

Line 331, remove double period.

Line 336, 364 and elsewhere. Please do not use „didn’t” or „don’t” but „did not” or „do not” instead. Please check this throughout the manuscript.

Line 404 research is recommended

Experimental design

See general comments above.

Validity of the findings

See general comments above.

Additional comments

See general comments above.

---

## Round 0.3 · Minor Revisions

· Academic Editor

Minor Revisions

Dear authors

Thank you for your revision.

In my final checks of your manuscript I have noticed the following:

Fig. 3 is not optimal.

A reader cannot see, which sequences are Vipera berus. Thus include V.b. before the location

Also: you provide a supspecues "sachaliensis" in one instance but not for the others; for sake of consistency, provide suspecies level also for the other samples, if possible.

Regards

M Wink
Academic editor

---

## Round 0.4 · accepted · Accept

· Academic Editor

Accept

Dear authors

Thanks for the revision. Thank you for submitting the ms

Michael Wink